# A Study on the Underwater Energy Harvester with Two PVDFs Installed on the FTEH and CTEH at the End of the Support

**DOI:** 10.3390/s23020808

**Published:** 2023-01-10

**Authors:** Jongkil Lee, Jinhyo An, Chonghyun Lee, Yoonsang Jeong, Hee-Seon Seo, Yohan Cho

**Affiliations:** 1Mechanical Engineering Education, Andong National University, Andong 36729, Republic of Korea; 2Ocean System Engineering, Cheju National University, Jeju 63243, Republic of Korea; 3Agency for Defense Development, Changwon 51682, Republic of Korea

**Keywords:** funnel-type energy harvester, cymbal-type energy harvester, cantilever type PVDF, vortex-induced vibration, voltage doubler rectifier

## Abstract

In this study, two thin rectangular PVDFs were installed in the form of a cantilever on a FTEH (funnel-type energy harvester), and a CTEH (cymbal-type energy harvester) was fabricated in a form coupled to the upper part of the support. As a result of measuring the energy harvesting sensitivity according to the installation direction of the CTEH, a high voltage was measured in the structure installed on top of the support across all flow velocity conditions. A composite structure PVDF energy harvester combining CTEH and FTEH was fabricated and the amount of power generated was measured. As a result of measuring the open-circuit voltage of the PVDF energy harvester device with a composite structure to which the optimum resistance of CTEH of 241 kΩ and the optimum resistance of FTEH of 1474 kΩ were applied at a flow rate of 0.25 m/s, the output voltage compared to the RMS average value was 7 to 8.5 times higher for FTEH than for CTEH. When the flow rate was 0.5 m/s, the electrical energy charged for 500 s was measured as 2.0 μWs to 2.5 μWs, and when the flow speed was 0.75 m/s, it reached 2.5 μWs when charged for 300 s, generating the same amount when the flow rate increased by 50%. The time to do it was reduced by 66.7%.

## 1. Introduction

The underwater acoustic sensor needs a semi-permanent and stable power supply to transmit and receive sound signals to detect sound. For this purpose, the need to develop an underwater energy harvester has emerged. Energy harvesters using fluid flow in water mainly harvest energy using the vibration of piezoelectric materials, and fluid passing around a cylinder or cable underwater causes vortex-induced vibration, and energy harvesting devices using this have been studied [1,2,3,4,5,6,7]. Actual energy harvesting is generated by the vibration of the cantilever-type piezoelectric element installed at the end, and PVDF or the MFC (macro fiber composite) are mainly used as piezoelectric elements [2,7].

Recently, Lee et al. [1] devised a funnel-type energy harvester (FTEH) using polyvinylidene fluoride (PVDF) and verified the energy harvest of the FTEH according to the flow rate through experiments. They install one PVDF at the end of the funnel type, and the FTEH has the effect of speeding up the flow rate at the outlet side because the cross-sectional area of the fluid inlet side is wider than the cross-sectional area of the outlet side [1]. Mehmood et al. [3] calculated the energy generated from the piezoelectric material according to the Reynolds number by using the vortex flow generated from the fluid flowing vertically through the circular cylinder fixed to the cable. Dai et al. [4] studied a piezoelectric energy harvesting device using vortex-induced vibration and studied that the maximum power was harvested at a flow velocity of around 1.2 m/s, and Grouthier et al. [5] studied the optimal energy harvesting due to induced vortex vibration in a circular cable and proved the efficiency of an energy harvesting device using a long cable. Song et al. [6] studied energy harvesting by vortex vibration of a piezoelectric cantilever installed perpendicular to the circumferential direction of the cylinder. In previous studies [3,4,5,6], energy harvesting by vortex vibration of a fluid flowing around a cylindrical object was proposed, and piezoelectric materials were commonly used. However, Erturk et al. [7] proposed an energy harvesting device using MFC instead of using conventional piezoelectric materials.

Unlike an energy harvester installed in a cantilever shape around a cylinder, a cymbal-type energy harvester has a structure in which symmetrical top and bottom caps are combined with a disc-shaped piezoelectric material, and the piezoelectric coefficient is amplified by the structure of the cymbal. The metal end cap by the cavity of the cymbal plays the role of a mechanical transformer that can transform and amplify a part of the incident axial stress in the radial stress [8,9,10,11]. The cymbal-type piezoelectric element has a very high energy conversion rate and can generate a large displacement. Recently, a piezoelectric single crystal has attracted attention due to its excellent electromechanical response. Cymbal-shaped structures are advantageous for micro-energy harvesting because they can generate large strains in the plane direction against external forces. Bezanson et al. [8] published a study on SURVIVE (supply utilizing vortex-induced vibration energy), which is a structure in which a cantilever is attached to the surface of a piezoelectric transducer-type cymbal. When the cantilever vibrates due to fluid flow, vortex flow is created. Experimental results showed that at least 6 mW of energy was generated when the fluid velocity was 0.25 m/s. Kim et al. [10] proposed a cymbal-type piezoelectric transducer for energy harvesting and it showed an output of 38 mW. Shim et al. [11] proposed a new technique of an equivalent circuit to analyze the transmitting characteristics of a cymbal array.

Since the AC current cannot be used directly in batteries and DC power applications, DC conversion through a rectifier circuit is required [1]. The forward voltage drop of the diode in a low-voltage circuit is a loss that cannot be ignored. To overcome this shortcoming, a circuit adopting a voltage doubler rectifier (VDR) has been proposed [12,13].

In this study, in the FTEH model of Lee et al. [1], the PVDF is installed in parallel, and the energy harvesting characteristics according to the length, installation direction, and fluid velocity of the PVDF are experimentally observed for the type in which the piezoelectric cymbal is coupled to the end of the support. It is expected that the results of this study can be used as basic data for designing a stable power supply for underwater sensors.

## 2. Basic Numerical Simulations

The relationship between the vibration displacement *y*(*t*) and the voltage $v_{e}\left(t\right)$ is expressed by simultaneous differential Equations (1) and (2), and the generated voltage can be obtained by solving them. The electromechanically coupled ordinary differential equations in modal coordinates are [1,2]
(1)$$\frac{d^{2}y(t)}{dt^{2}}+2\delta \omega_{r}\frac{dy(t)}{dt}+\omega_{r}^{2}y\left(t\right)-\varepsilon_{r}v_{e}\left(t\right)=f_{r}(t)$$
(2)$$C_{p}\frac{dv_{e}\left(t\right)}{dt}+\frac{v_{e}\left(t\right)}{R}+\sum_{r=1}^{\infty }\varepsilon_{r}\frac{dy(t)}{dt}=0,$$
where

y(t) = transverse displacement at position *x* and time *t*;

$v_{e}\left(t\right)$ = voltage response across the external resistive load *R*;

$\omega_{r}$ = undamped natural frequency in constant electric field conditions;

*R* = external resistive load;

δ = modal mechanical damping ratio;

$\varepsilon_{r}$ = modal electromechanical coupling;

$C_{p}$ = depends on the way piezoceramic layers;

$f_{r}(t)$ = modal forcing function.

It can predict the coupled system dynamics and one obtains voltage response which depends on the vibration displacement [2]. In the numerical simulation, coefficients in Equations (1) and (2) assumed constant numbers, which means that all the components of FTEH were judged as fixed values. The transition of generated power by generated voltage was simulated. The source of excitation is a sinusoidal function $f_{r}(t)$ in Equation (1). Dimensionless power generated by the PVDF and its frequency spectrum, which are generated when all initial conditions are 0.5, are shown in Figure 1. As shown in Figure 1, it can be seen that as the vibration displacement of the PVDF increases, the amount of power generated increases in proportion. As shown in Figure 1 it can be seen that the trend of change in the amount of dimensionless power generation changes at a frequency of 2 Hz or less. As shown in Figure 1, it can be seen that the power generated in the PVDF changes according to the change in vibration displacement and voltage.

## 3. Experimental Results and Discussions

### 3.1. Configuration of Experimental Setup

In order to measure the energy harvest due to the fluid flow in the water experimentally, an experimental device was fabricated as shown in Figure 2. As shown in Figure 2, the experimental device is divided into an upper structure and a lower structure, an underwater energy harvester device is installed in the upper structure, and a submersible water pump and a storage tank for fluid circulation are installed in the lower structure. The fluid was circulated to the superstructure by a submersible pump and the fluid velocity was controlled using a flow control valve. The flow rate was automatically measured when the fluid passed through the underwater energy harvester passed the flow rate measuring device, and the flow rates used in the experiment were 0.25 m/s, 0.50 m/s, and 0.75 m/s. A wire mesh with fine holes was installed so that the fluid passing through the outlet of the pump flows at a constant speed.

### 3.2. Energy Harvester According to PVDF Length and Installation Direction

Lee et al. [1] used one PVDF piezoelectric film; it was first conceived to install two PVDFs in parallel in one harvester to increase the output power efficiency. The method of installing two PVDFs is classified into three types as shown in Figure 3. As shown in Figure 3a,b, the structure and size of the two PVDFs of each size connected in parallel are the same as (b), but the funnels are installed symmetrically on both sides of Figure 3c. It was made in three of the same shape.

In order to compare the output efficiency according to the PVDF installation direction and flow velocity (0.25, 0.50, 0.75 m/s), experiments were conducted by dividing the total into 18 cases. Here, the support for fixing and supporting the harvester under fixed conditions was used in a flexible type that shows high efficiency, and a spiral-structured harvester model that increases vortex generation at the inlet of the harvester was used.

The open-circuit voltages of two PVDFs were measured simultaneously using an NI voltmeter, and the RMS voltage values for each flow rate were compared by averaging the measurement results of three harvesters for each case, as shown in Figure 4. As shown in Figure 4, the highest RMS voltage of 171.1 mV was shown in the one-sided funnel type (Case A, AV) when the flow velocity was 0.75 m/s. It was found that the vertical direction represents the maximum voltage in all experiments except for the one-sided funnel (Case A) at a flow velocity of 0.5 m/s. In the case of the merged structure harvester placed horizontally on the water surface during the experiment, it was confirmed that the two PVDFs overlap each other in a downward drooping state due to the influence of gravity, limiting the vibrational displacement of each PVDF. For this reason, it can be interpreted that the measured open-circuit voltage shows higher output in the vertical direction than in the horizontal direction. Comparing the one-sided funnel type of Case B and the two-sided funnel type of Case C, it can be seen that the output voltage of Case B is higher than that of Case C overall. However, when the flow velocity is 0.25 m/s, Case C has a higher output voltage than Case C. Considering the size of the PVDF piezoelectric, Case C installed in the vertical direction is more effective than Case A, which shows the maximum voltage.

The method of fixing the harvester of the PVDF combined structure to the support is largely classified into two types as shown in Figure 5. Figure 5a is a model in which rotational freedom is constrained in the direction of the support axis, and the rotation variable of the harvester itself is removed to compare the output according to the shape of the harvester. So far, the output results were compared using the integral fixture throughout the experiment, but in this experiment, the output efficiency according to the axial rotation variable was verified by additionally using the separate fixture as shown in Figure 5b.

To verify the output efficiency according to the axial rotation variable, the flow velocity was fixed at 0.25 m/s and the harvester model used a PVDF model with a spiral structure installed at the inlet on flexible support. As for the flow condition, the experiment was performed by selecting the PVDF installation direction and whether horizontal axis rotation was possible, and the experiment environment and classification of the experiment are shown in Figure 6 and Table 1, respectively.

Six experiments were performed from A to F, and the open-circuit voltage of PVDF no. 1 and no. 2 and the current of PVDF no. 1 were measured simultaneously using a 2-Port oscilloscope. For the rectified capacitor charging voltage, the rectified currents of PVDF no. 1 and no. 2 were measured simultaneously using a rectifier circuit composed of the same capacitor and diode. As shown in Figure 7, high power was output in the vertical direction when (a) the integral model was used, and (b) high power was output in the horizontal direction when the detachable model was used. The maximum power output was 0.154 V, 24.382 nW, which was shown in Case F (axial rotation fixed, PVDF installed horizontally).

### 3.3. Cymbal-Type Energy Harvester (CTEH) Performance Test

The CTEH has a structure in which two piezoelectric materials face each other in a curved state and uses the principle that voltage is generated when pressure is applied from the outside. Most of the CTEHs are mainly used as transducers [11], but in this study, they were used as energy-harvesting devices, as shown in Figure 8a. The shape of the CTEH is circular, with two protruding wires for output, so a jig to connect to the support was required, semi-circular and circular support frames were manufactured, and the CTEH was fixed as shown in Figure 8a. As shown in Figure 8, a symbol-type harvester with a symbol-type sensor built into the bottom of the supporter was fabricated and its performance was tested.

Since the surface of CTEH is made of a polymer coating layer, it was determined that the shape it had would change depending on the method of bonding with the end of the support, so type_B and type_C were combined in two ways. That is, in Figure 8b, type_B (B1, B2, BB), in which the plastic support is adhesively fixed on the upper surface of the CTEH polymer coating, and type_C (C1, C2), in which the support is directly adhesively and fixed to the metal surface after removing the CTEH coating. Open-circuit voltages in a total of five cases were measured and compared. In the case of type_B and type_C, in order to compare the performance according to the bonding and fixing methods, each case was manufactured with different bonding and fixing methods. As a fixed condition, the flow velocity was 0.25 m/s, and the output voltage due to the magnetic vibration of the support itself was measured, and the output voltage was measured with the funnel-type energy harvester installed as shown in Figure 8c. The effect on the performance of the harvester was identified and the experimental results are shown in Figure 9 and Figure 10.

Figure 9 shows the minimum, maximum, and rms values(dot line) of the output voltage when the CTEH is installed alone and when the FTEH is combined in the middle of the support. At this time, FTEH was only installed, and the output voltage was not measured, only the output voltage at CTEH was measured. As shown in Figure 10, when the plastic support was adhesively fixed on the upper surface of the CTEH polymer coating and CTEH and FTEH were installed at the same time, the RMS average voltage of the output voltage was 0.019 V, showing the highest output. In the case of the support, compared to the output value by magnetic vibration of the support itself, when the energy harvester device (FTEH) was installed, the RMS average voltage showed a performance improvement of 1.3 to 1.8 times that of other supporters.

When measuring energy harvesting by applying CTEH, it is necessary to derive the optimal resistance because the output voltage value varies depending on the resistance value used by the output terminal. As shown in Figure 11, in the structure where the CTEH is installed at the bottom of the support, the flow velocity was fixed at 0.25 m/s and the resistance value was varied to find the optimum resistance. As shown in Figure 11b, the maximum amount of power of 0.052 nW was generated at the resistance value of 241 kΩ. Therefore, the optimum resistance of CTEH is determined to be 241 kΩ.

The open-circuit voltage of the PVDF piezoelectric film and the open-circuit voltage of CTEH were measured at a constant flow rate of 0.25 m/s as shown in Figure 11 by applying the optimal resistance value of 241 kΩ. As shown in Figure 12 and Figure 13, the PVDF open-circuit voltage versus the RMS value increased by 40% from 0.082 V to 0.113 V, respectively, when the CTEH supporter type B2 was used, compared to when the conventional type A support was used. The additionally obtained open-circuit voltage was 0.014 V, accounting for 11.1% of the output voltage of the type B energy harvester.

The overall shape of the CTEH is a structure in which circular plates are stacked up and down, and the surface is covered with polyurethane. CTEH is symmetrical on the upper and lower surfaces, so there is no upper and lower division, but it is judged that there will be a difference when the pressure fluctuation (due to the fluid flow) acts in a direction parallel to the surface of the circular plate and when it acts perpendicularly. In order to see the energy harvesting sensitivity according to the installation direction of the CTEH, as shown in Figure 14, the structure in which the CTEH is installed below the support (CTEH_lower, Figure 14b) and the structure installed above the support (CTEH_upper, Figure 14c) made by dividing. Figure 14b is a structure in which pressure is applied parallel to the fluid flow, and Figure 14c is a structure in which pressure is received perpendicularly to the fluid flow. It was judged that the output signal would be higher when the CTEH was positioned in the direction perpendicular to the flow of the fluid, so the position of the CTEH was moved from the floor to the end of the support and the measurement results were compared. As shown in Figure 14, the sensitivity experiment was conducted according to the installation direction of the CTEH by measuring the open-circuit voltage according to the wind speed of three types of energy harvesters, FTEH, CTEH_lower, and CTEH_upper, in a laboratory environment. In general, experiments underwater are more complicated than in the air. Therefore, an experiment was conducted in the air to quickly find out the change in voltage generation according to the attachment position of the CTEH.

The flow velocity was divided into 0.1 m/s, 0.25 m/s, and 0.7 m/s and tested. Figure 15 and Figure 16 show the measured voltage generated by blowing air with the CTEH installed for the three models shown in Figure 14. As a result of the sensitivity measurement, high voltages were measured in the order of FTEH > CTEH_upper > CTEH_lower across all flow conditions. The open voltage compared to the RMS average value based on the flow velocity of 0.25 m/s showed 20 to 30 times the output value of FTEH compared to CTEH, and CTEH occurred when installed at the top of the support, which is perpendicular to the flow, rather than buried at the bottom of the support.

It can be seen that the voltage is high in the case of FTEH alone. In all three cases of Figure 14, the generated voltage increased as the flow rate increased. Therefore, when fabricating the composite structure PVDF energy harvester, the CTEH was fabricated to be located on the top of the support using the results in Figure 16. Since the output voltage of the energy harvesting device is a time-variant quantity depending on the instability of fluid flow and changes in the experimental environment, a more stable rms voltage was used for data analysis.

### 3.4. Composite Structure PVDF Energy Harvester Performance Test

A composite structure PVDF energy harvester combining CTEH and FTEH was fabricated as shown in Figure 17 and performance tests were performed. The PVDF piezoelectric film in the form of a thin film is watertight, installed in multiple layers, and a spiral-shaped vortex generator is installed at the inlet of the fluid inlet, and at the same time, the acrylic guide is manufactured in a funnel type that narrows from the inlet to the outlet, so that the flow rate at the outlet is reduced. An energy harvester with this increasing form was devised. Reflecting on the results of the sensitivity test of CTEH, finally, a composite structure PVDF energy harvester device with CTEH installed on the top of the support was fabricated. As shown in Figure 17, the composite structure PVDF energy harvester is largely composed of FTEH, CTEH, and flexible support.

The FTEH data measured in the fabricated water tank environment are open-circuit voltage data and rectified output data. The data were measured according to the environmental conditions and manufacturing conditions of FTEH. In the case of open-circuit voltage measurement, it was measured equally across all environmental conditions and manufacturing conditions, and the equivalent model of the measurement system is shown in Figure 18.

As a result of measuring the open-circuit voltage of the composite structure PVDF energy harvester device to which the optimum resistance of CTEH was 241 kΩ and the optimum resistance of FTEH was 1474 kΩ, and the flow rate was 0.25 m/s, the output voltage compared to the RMS average value was 7 to 8.5 times higher for FTEH than for CTEH. The open-circuit voltage test results are shown in Figure 19 and Figure 20, respectively. In Figure 19 and Figure 20, the left and right PVDFs of FTEH are expressed as PVDF_L and PVDF_R, respectively, and CTEH is expressed as a cymbal. As shown in Figure 20, the open-circuit voltage (RMS) values of CTEH were measured as 0.0088 V, PVDF_L as 0.061 V, and PVDF_R as 0.074 V. It was found that the voltage generated at FTEH was higher than that generated at CTEH when the flow velocity was low.

The charging voltage and charging energy of the rectified capacitor of the composite PVDF energy harvester were measured according to the flow velocity conditions of 0.25 m/s, 0.50 m/s, and 0.75 m/s, respectively, and are shown in Figure 21. The amount of charging energy of CTEH was negligibly smaller than that of FTEH, and the resulting values were summed by independently charging the rectifier circuit for PVDF at each flow rate condition. As shown in Figure 21, the time to reach the same stored energy of harvesting according to the flow rate was measured, and when the reference value was set at 1.5 µW s, it was confirmed that the arrival time took 39 min, 375 s, and 225 s, respectively. It was confirmed that the electric energy storage of the composite structure energy harvester in which CTEH and FTEH were combined increased rapidly as the flow rate increased, and it was found that the FTEH structure had more electric energy storage than the CTEH structure.

## 4. Conclusions

In this study, a device that harvests electric energy using an underwater fluid flow was devised and its performance was tested. FTEH is a form in which a thin film PVDF piezoelectric film is watertight, installed in multiple layers, and a spiral-shaped vortex generator is installed at the inlet of the fluid inlet, and at the same time, the acrylic guide becomes narrower from the inlet to the outlet. That is, two thin rectangular PVDFs were installed in a cantilever shape at the end of the FTEH. The proposed energy harvester was manufactured in the form of a combination of a CTEH, which is a method in which the FTEH and the symbol are symmetrically combined up and down to apply pressure to the piezoelectric material in the center, and the change in electric energy charge according to the change in flow rate measured. To see the difference in generated voltage according to the PVDF installation direction, the open-circuit voltage of two PVDFs was measured simultaneously. The measured open circuit voltage showed higher output in the vertical direction than in the horizontal direction. As a method of fixing the FTEH to the support, the output results were compared using a model in which the FTEH rotates freely in the direction of the axis of the support and a model in which the degree of freedom of rotation is constrained.

The surface of CTEH is made of a polymer coating layer, so it is assumed that the shape it has will change depending on how it is combined with the end of the support. After that, the open circuit voltage of the type in which the support was directly adhesively fixed to the metal surface was measured and compared. When the plastic support was adhesively fixed on the upper surface of the CTEH polymer coating and CTEH and FTEH were installed at the same time, the RMS average voltage of the output voltage showed the highest output with 0.019 V. When measuring energy harvesting by applying CTEH, it is necessary to derive the optimal resistance, so the optimal resistance was found by changing the resistance value by fixing the flow speed at 0.25 m/s in the structure where the CTEH is installed at the bottom of the support. The optimum resistance was 241 kΩ, resulting in a maximum power consumption of 0.052 nW. In order to examine the energy harvesting sensitivity according to the installation direction of the CTEH, a sensitivity test was performed by measuring the open circuit voltage according to the wind speed by dividing the CTEH into a structure installed below the support and a structure installed above the support. As a result of the sensitivity measurement, a high voltage was measured in the structure installed on top of the support across all flow velocity conditions. That is, it was found that the voltage generated when CTEH was installed at the upper end of the support in the direction perpendicular to the flow rather than being buried at the lower end of the support.

A composite structure PVDF energy harvester combining a CTEH and a FTEH was fabricated and the amount of power generated was measured. Reflecting on the results of the sensitivity test of CTEH, a composite structural energy harvester device with CTEH installed on the top of the support was finally fabricated. The FTEH data measured in the fabricated water tank environment are voltage and rectified output data. In the case of voltage measurement, it was measured equally across all environmental conditions and manufacturing conditions, and an equivalent model of the measurement system was also presented. The harvester’s rectifier circuit output was obtained by measuring the charging voltage of the rectifier capacitor for the condition having the high RMS voltage value among the measured voltage data of various conditions. Basically, a bridge rectifier circuit was used when comparing the charge amount according to the harvester condition, and finally, a voltage doubler rectifier circuit with the highest storage efficiency among the three rectifier systems was used to measure the rectified voltage to achieve the target amount.

As a result of measuring the open-circuit voltage of the PVDF energy harvester device with a composite structure to which the optimum resistance of CTEH of 241 kΩ and the optimum resistance of FTEH of 1474 kΩ were applied at a flow rate of 0.25 m/s, the output voltage compared to the RMS average value was 7 to 8.5 times higher for FTEH than for CTEH. appear. It was found that the voltage generated by FTEH was higher than the voltage generated by CTEH when the flow rate was low. The charging voltage and charging energy of the rectifier capacitor of the composite structure PVDF energy harvester were measured, and the charging energy of CTEH was negligibly smaller than that of FTEH. When the flow rate is 0.5 m/s, the electrical energy charged for 500 s was measured as 2.0 to 2.5, and when the flow rate was 0.75 m/s, it reached 2.5 when charged for 300 s, generating the same amount when the flow rate increased by 50%. The time to do it was reduced by 66.7%.

In order to increase the amount of power generation, FTEH and CTEH can be connected in series to fully utilize vortex-induced vibration. The FTEH and CTEH devices used in this study will greatly contribute to the stable power supply of underwater acoustic sensors used to detect objects in the ocean environment.

## Figures and Tables

**Figure 1 sensors-23-00808-f001:**
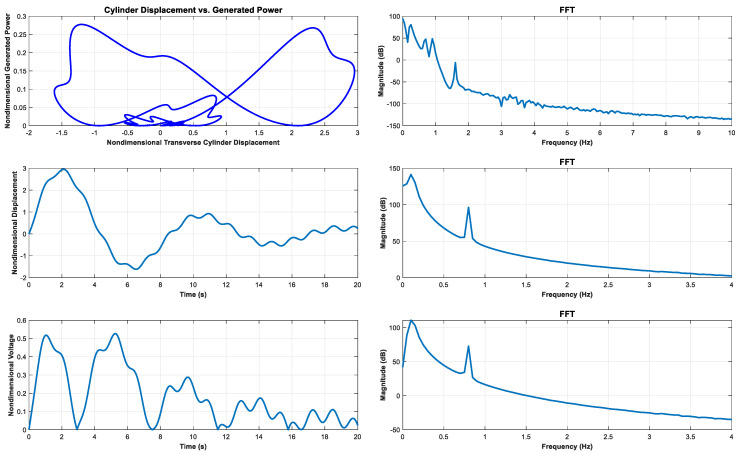
Generated power, nondimensional displacement, nondimensional voltage, and their frequency spectrum, respectively.

**Figure 2 sensors-23-00808-f002:**
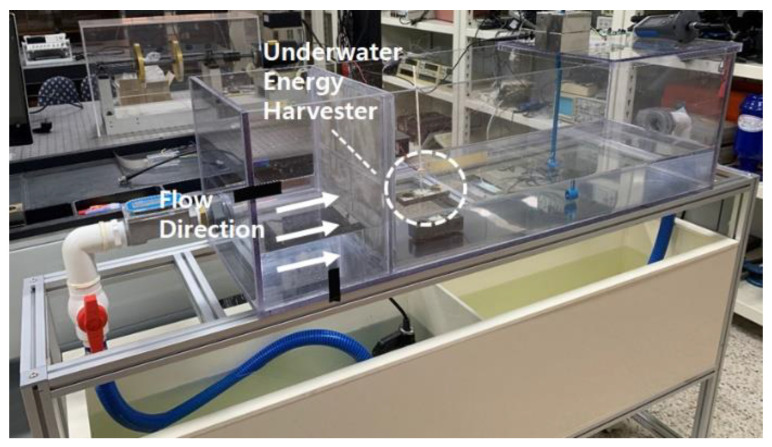
Experimental setup.

**Figure 3 sensors-23-00808-f003:**
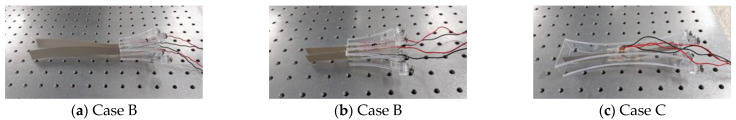
2 PVDF combined structure harvesters ((**a**) Case A: PVDF size 155.7 × 18 (mm^2^) with 2 mm thickness one-sided funnel (inlet = 58 mm × 25 mm, outlet = 29 mm × 25 mm, installation height from the base = 140 mm) type. (**b**) Case B: PVDF size 61.47 × 12 (mm^2^) one-sided funnel type. (**c**) Case C: PVDF size 61.47 × 12 (mm^2^) both-sided funnel types).

**Figure 4 sensors-23-00808-f004:**
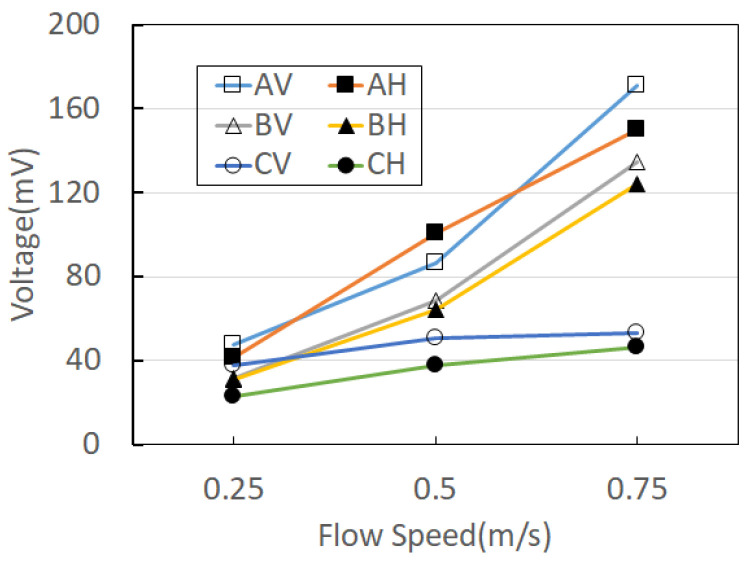
Two PVDF-combined structure harvesters (V is vertical, H is horizontal of the types A, B, and C, respectively).

**Figure 5 sensors-23-00808-f005:**
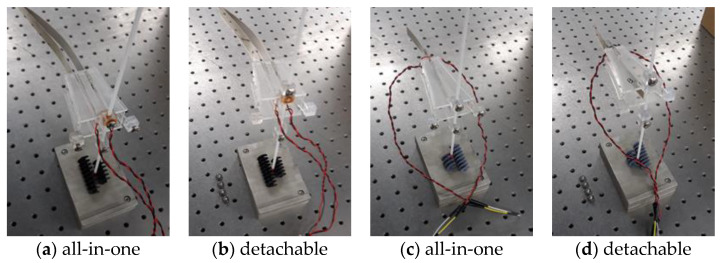
(**a**,**b**); One-sided funnel (Long/out), (**c**,**d**); Two-sided funnel (Short/in).

**Figure 6 sensors-23-00808-f006:**
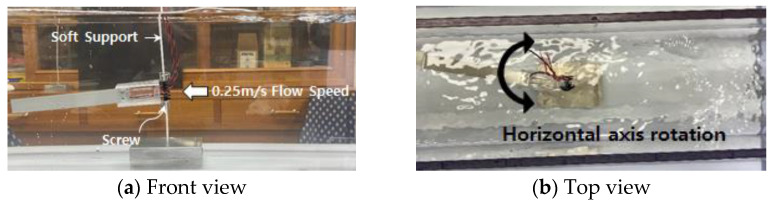
Power efficiency verification model according to axial rotation variable.

**Figure 7 sensors-23-00808-f007:**
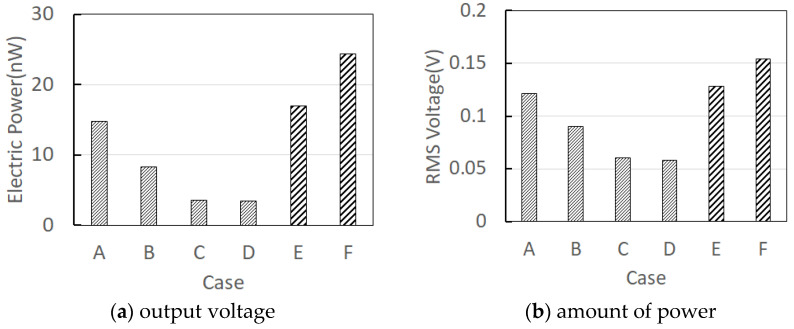
Comparison of output voltage and power consumption according to axial rotation conditions of case A~F as shown in the Table 1.

**Figure 8 sensors-23-00808-f008:**
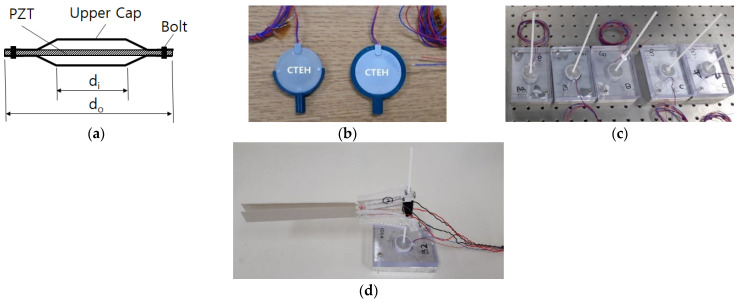
CTEH mounting location and experimental setup combined with FTEH. ((**a**) Schematic diagram of the CTEH, (**b**) fabricated CTEH, (**c**) CTEH installed at the bottom of the support (case of B1, B2, BB, C1, C2), (**d**) combination of CTEH and FTEH installed at the bottom of the support).

**Figure 9 sensors-23-00808-f009:**
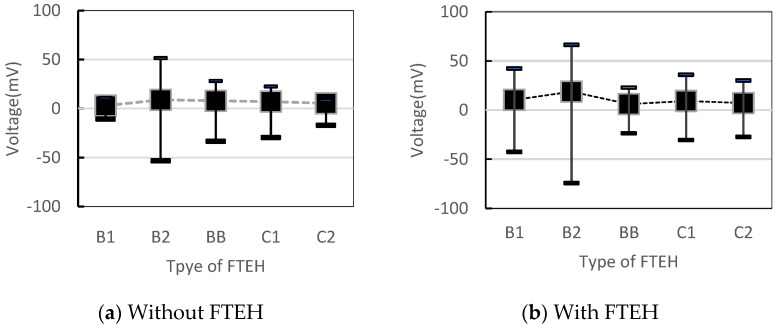
Comparison of output voltages depending on how the CTEH is bonded to the support.

**Figure 10 sensors-23-00808-f010:**
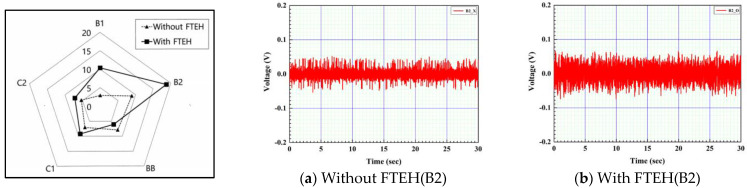
Comparison of output voltage according to the coupling method of CTEH installed at the bottom of the support.

**Figure 11 sensors-23-00808-f011:**
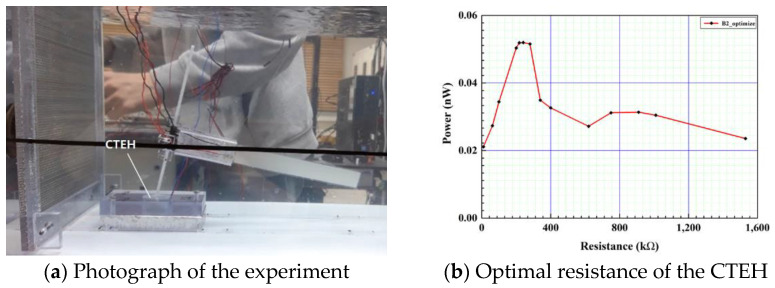
CTEH optimal resistance measurement and results.

**Figure 12 sensors-23-00808-f012:**
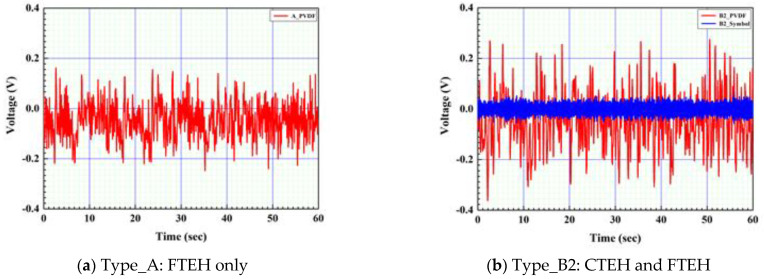
Energy harvester open voltage measurement results.

**Figure 13 sensors-23-00808-f013:**
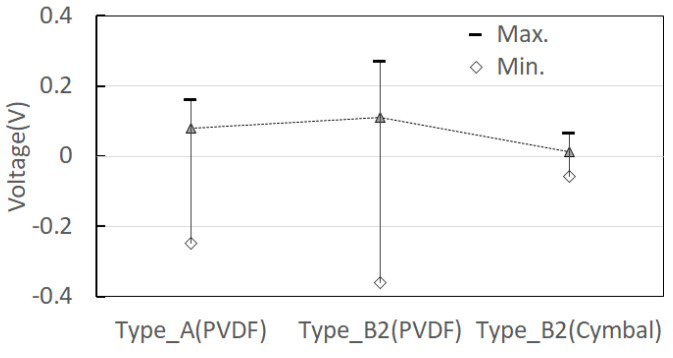
Energy harvester open voltage measurement with optimum resistance.

**Figure 14 sensors-23-00808-f014:**
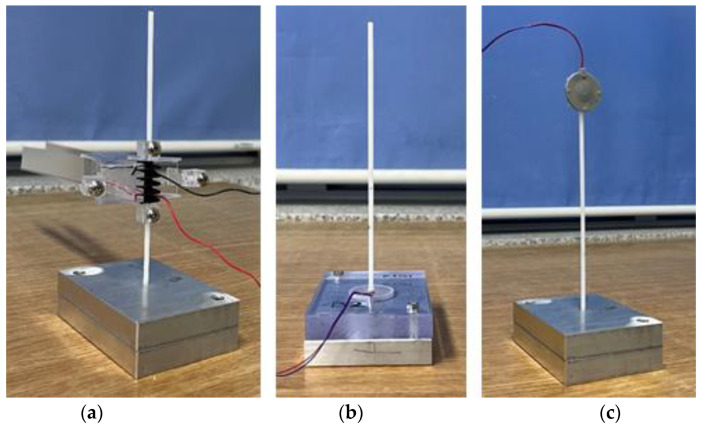
Energy harvester sensitivity experimental model ((**a**) FTEH, (**b**) CTEH lower, (**c**) CTEH upper, respectively).

**Figure 15 sensors-23-00808-f015:**
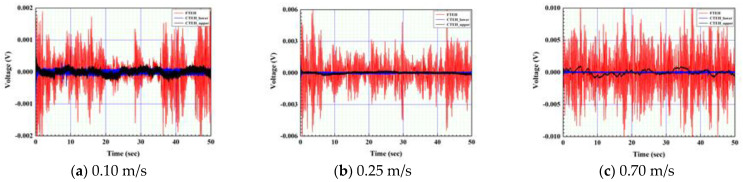
Energy harvester sensitivity test results by flow rate.

**Figure 16 sensors-23-00808-f016:**
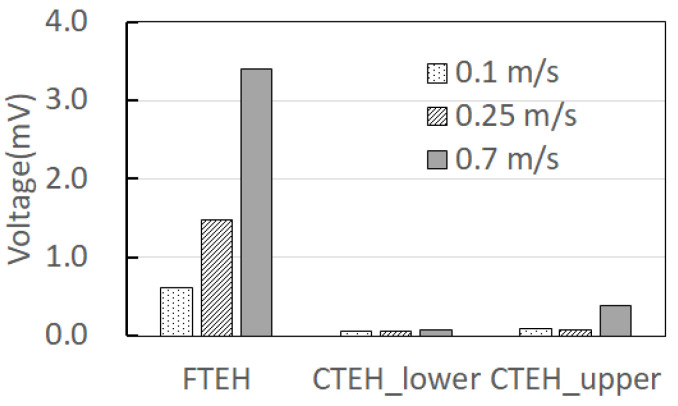
Comparison of energy harvester sensitivity test results by flow rate.

**Figure 17 sensors-23-00808-f017:**
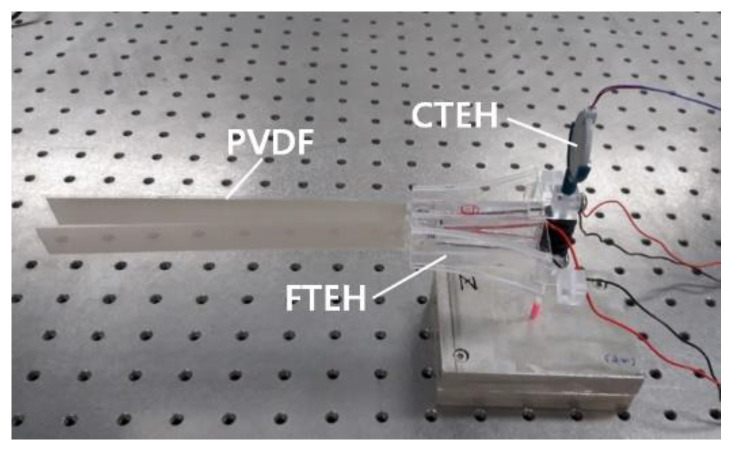
Composite structure PVDF energy harvester.

**Figure 18 sensors-23-00808-f018:**
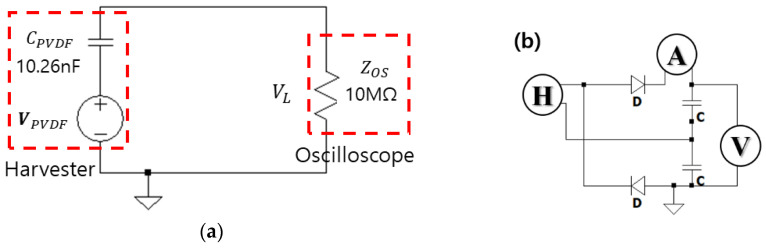
(**a**) Equivalent model of the open-circuit voltage measurement system and (**b**) rectifier circuit measurement system (schematic diagram).

**Figure 19 sensors-23-00808-f019:**
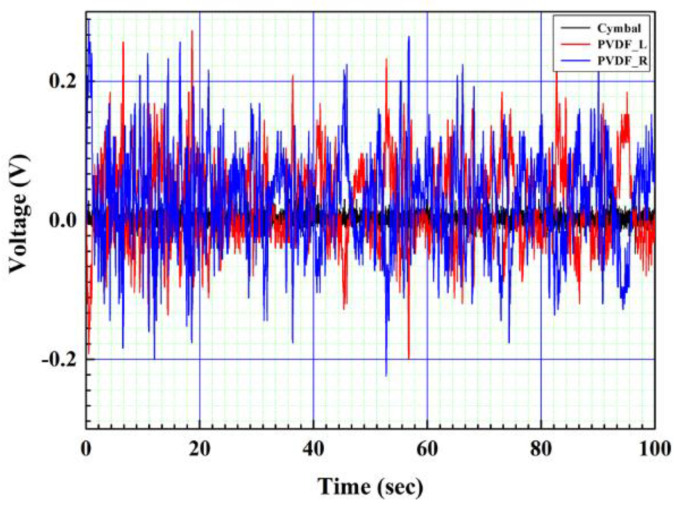
Composite structure PVDF energy harvester real-time voltage characteristics.

**Figure 20 sensors-23-00808-f020:**
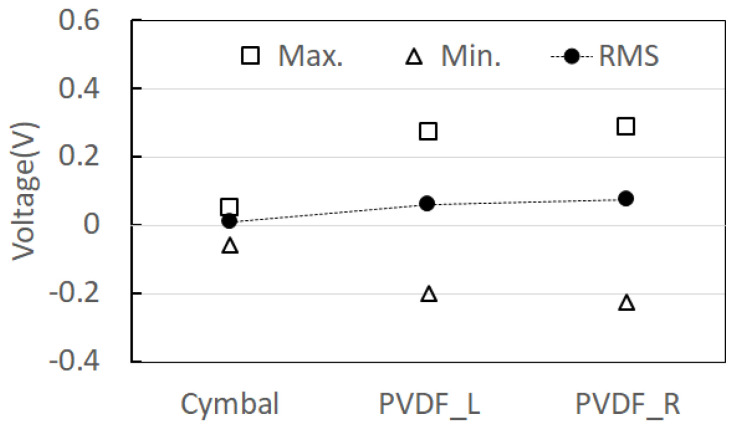
Composite structure PVDF energy harvester open voltage measurement result.

**Figure 21 sensors-23-00808-f021:**
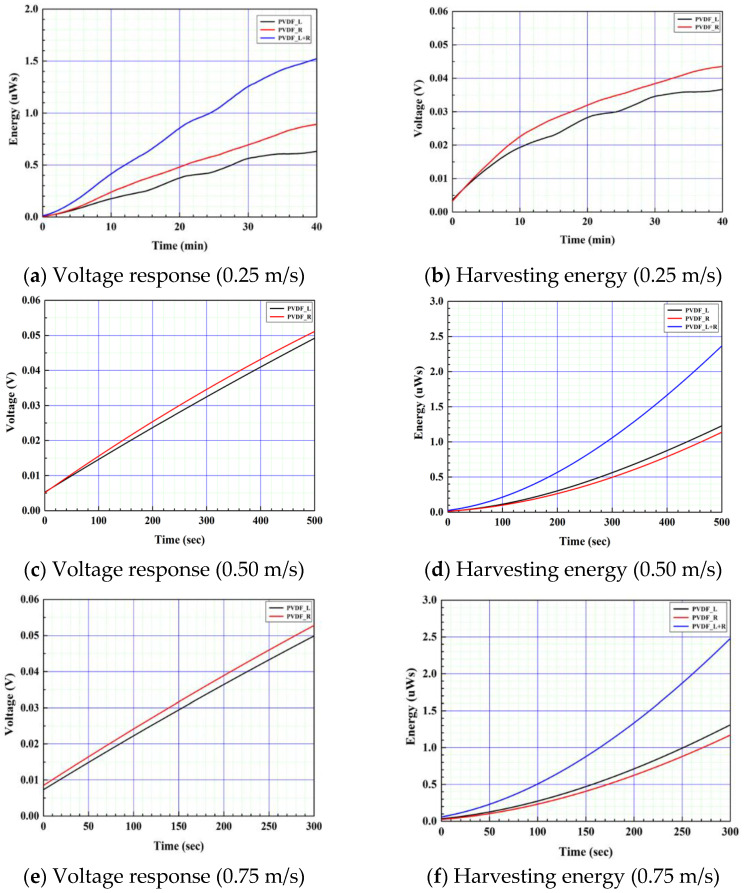
Composite structure PVDF energy harvester rectifier capacitor charging test results under the various flow speed of 0.25 m/s, 0.50 m/s, and 0.75 m/s, respectively.

**Table 1 sensors-23-00808-t001:** Power efficiency verification model according to axial rotation variable.

Horizontal Axis Rotation	Electrode Size	Direction	Case
Free Rotation	155.7 mm × 18 mm	Vertical	A
		Horizontal	B
	61.47 mm × 12 mm	Vertical	C
		Horizontal	D
Fixed	155.7 mm × 18 mm	Vertical	E
		Horizontal	F

## Data Availability

Not applicable.

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
