# Peer review of "A Study on the Underwater Energy Harvester with Two PVDFs Installed on the FTEH and CTEH at the End of the Support"

_sensors, 2023, doi:10.3390/s23020808_

Round 1

Reviewer 1 Report

In this work, the design of the underwater energy harvester is innovative. Also, the tank tests of this energy harvester were done and the results were analyzed. I think the results are very useful and interesting.

Some minor problems and modification suggestion are as follows.

1 This study lacks some theoretical depth.

2 This paper was well written and easy to follow. But the coordinates in the figures had different font sizes. Including fig.1. ,fig.9., fig.10., fig.11. etc

3 Add some more specific application scenarios.

This work was focused on equipment and experimental research. Even without some theoretical innovation, I still think the results are interesting and this work can be published after minor revisions.

Author Response

Thanks to the reviewers for their good comments. Based on the reviewer's comments, the contents of the paper have been revised and supplemented as follows. Please find attached pdf file and revised parts of the paper are marked in red font.

1 This study lacks some theoretical depth.

  • In page 2, 3 theoretical approach has been modified as follows. "
  • The relationship between the vibration displacement y(t) and the voltage  is expressed by simultaneous differential equations (1) and (2), and the generated voltage can be obtained by solving them. The electromechanically coupled ordinary differential equations in modal coordinates are [1, 2]

see eqs. (1) and (2) in the attached pdf file.

  = transverse displacement at position x and time t;

   = voltage response across the external resistive load R;

  =undamped natural frequency in constant electric field conditions;

R=external resistive load;

= modal mechanical damping ratio;

= modal electromechanical coupling;

=depends on the way piezoceramic layers;

= modal forcing function.

2. This paper was well written and easy to follow. But the coordinates in the figures had different font sizes. Including fig.1. ,fig.9., fig.10., fig.11. etc

  • Font size of the figs 1, 10, 11 are came from measurement or common software, therefore it is very hard to change font. But the font size in fig. 9 has been changed.

3. Add some more specific application scenarios.

  • In conclusions we added this statement. "In order to increase the amount of power generation, FTEH and CTEH can be connected in series to fully utilize vortex induced vibration. The FTEH and CTEH devices used in this study will greatly contribute to the stable power supply of underwater acoustic sensors used to detect objects in the ocean environment."

Thanks again to the reviewers for their good comments.

Reviewer 2 Report

This paper is potentially interesting and includes a lot of experimental results, nevertheless some improvements must be made:

1)      Some clear schemes of the harvesters are needed, from the photos the geometry and sizes of the devices cannot be understood

2)      More details about equations 1 and 2 must be given and the meaning of all symbols must be given.

3)      What is the source of excitation in numerical simulations?

4)      In the descriptions of figure 10 minimum, maximum and RMS voltage are mentioned, but I see only one line.

5)      At page 10 tests with air are mentioned. The proposed device will be used underwater, why air tests are carried out? Please clarify.

6)      The difference between output voltage and RMS voltage must be clarified, typically the RMS value is a figure that defines a time-variant quantity.

Author Response

Thanks to the reviewers for their good comments. Based on the reviewer's comments, the paper was revised and supplemented as follows.

1. Some clear schemes of the harvesters are needed, from the photos the geometry and sizes of the devices cannot be understood.  

  • revised in page 4:  "In figure 3, PVDF combined structure harvesters((a) Case A: PVDF size 155.7 x 18 (mm2) with 2mm thickness one-sided funnel(inlet=58mm x 25mm, outlet=29mm x 25mm, installation height from the base=140mm) type" In table 1, unit of mm inserted.

2. More details about equations 1 and 2 must be given and the meaning of all symbols must be given.

  • revised in page 2, 3: "The relationship between the vibration displacement y(t) and the voltage  is expressed by simultaneous differential equations (1) and (2), and the generated voltage can be obtained by solving them. The electromechanically coupled ordinary differential equations in modal coordinates are [1, 2]

See the attached pdf file

where

  = transverse displacement at position x and time t;

   = voltage response across the external resistive load R;

  =undamped natural frequency in constant electric field conditions;

R=external resistive load;

= modal mechanical damping ratio;

= modal electromechanical coupling;

=depends on the way piezoceramic layers;

= modal forcing function."

3. What is the source of excitation in numerical simulations?

  • revised in page 3: "The source of excitation is sinusoidal function  in eq. (1). "

4.    In the descriptions of figure 10 minimum, maximum and RMS voltage are mentioned, but I see only one line.

  • revised in page 8: "Figures 9 shows the minimum, maximum, and rms values(dot line) of the output voltage when the CTEH is installed alone and when the FTEH is combined in the middle of the support. "

5.  At page 10 tests with air are mentioned. The proposed device will be used underwater, why air tests are carried out? Please clarify.

  • revised in page 10: "In general, experiments in underwater are more complicate than in air. Therefore, an experiment was conducted in the air to quickly find out the change in voltage generation according to the attachment position of the CTEH."

6. The difference between output voltage and RMS voltage must be clarified, typically the RMS value is a figure that defines a time-variant quantity.

  • revised in page 11: "Since the output voltage of the energy harvesting device is a time variant quantity depending on the instability of fluid flow and changes in the experimental environment, a more stable rms voltage was used for data analysis."

 Thanks again to the reviewers for their good comments.

Round 2

Reviewer 2 Report

The paper was improved according to my suggestions, now it can be published.